# Experimental Research on Magnesium Phosphate Cements Modified by Fly Ash and Metakaolin

**He Liu** [1,2,*]**, Qidong Feng** [1]**, Yanhai Yang** [1]**, Jingyi Zhang** [3,*]**, Jian Zhang** [4] **and Guangchao Duan** [1]

1   School of Transportation and Geometics Engineering, Shenyang Jianzhu University, Shenyang 110168, China; fengqidong666@163.com (Q.F.); yangyanhai168@126.com (Y.Y.); d1045339052@163.com (G.D.)
2   National Engineering Research Center of Highway Maintenance Technology, Changsha University of Science and Technology, Changsha 410114, China
3   School of Civil Engineering, Shenyang Urban Construction University, Shenyang 110167, China
4   Liaoning Datong Highway Engineering Co., Ltd., Shenyang 110005, China; zhangjian613935@163.com
*   Correspondence: heliu@sjzu.edu.cn (H.L.); dq_zjy@syucu.edu.cn (J.Z.)

**Abstract:** To increase performance and save costs when utilizing magnesium phosphate cements (MPC) to repair a damaged building structure or a cement pavement, MPC is typically combined with fly ash (FA) and metakaolin (MK). The influence of FA and MK on the workability, rheological characteristics, flexural strength, compressive strength, and drying shrinkage of MPC was investigated in this research. MPC samples with different percentages of FA and MK by weight replacement were prepared. The results indicate that an appropriate dosage of MK and FA could decrease MPC fluidity and delay the setting time. MPC's yield stress and plastic viscosity were increased when MK was added. FA has a negative influence on flexural and compressive strength as compared to control MPC and the compressive strength of MPC with MK increases and then decreases. The drying shrinkage of MPC containing MK and FA is superior to control mixture. MPC with 10% FA and 10% MK has the best-modified performance in terms of the comprehensive performance of MPC at all test ages.

**Keywords:** magnesium phosphate cement; metakaolin; fly ash; workability; rheological properties; drying shrinkage





## 1. Introduction

Magnesium phosphate cement (MPC) is a distinctive inorganic material in which the acid-base reaction is usually used as a reasonable explanation for its hydration process [1–4]. MPC materials are known for their quick hardening, excellent early strength, and durability. They can be used for a variety of things, including roadway repair, bridge decking, and airport runways [5–7]. The commonly used MgO is dead burned magnesium oxide, and the phosphate salts are mainly ammonium dihydrogen phosphate ($NH_4H_2PO_4$, ADP) and potassium dihydrogen phosphate ($KH_2PO_4$, PDP). The characteristics such as specific surface area and fineness of MgO could significantly influence the hydration rate and mechanical properties of MPC. The main hydration product is struvite ($MgNH_4PO_4.6H_2O$), and the chemical equation can be expressed as follows:

$$MgO + NH_4H_2PO_4 + 5H_2O \rightarrow MgNH_4PO_4 \cdot 6H_2O$$

$$MgO + KH_2PO_4 + 5H_2O \rightarrow MgKPO_4 \cdot 6H_2O$$

The mix proportion of an MPC paste is determined by two main factors: the magnesia-to-phosphate molar ratio (M/P) and the water-to-cement mass ratio (W/C). M/P plays a decisive role on the hydration rate, setting time, and strength development of MPC. Previous studies demonstrated that MPC material had the best mechanical properties due to its optimal M/P. Excessive phosphate wasn't conducive to the development of mechanical

strength [8–10]. Lower W/C was usually determined to produce MPC pastes which had robust performance and great durability for the best design of MPC cements [10–12]. In addition, the hydration temperature also affects the strength development of MPC. In the early hydration stage of MPC, the hydration is very fast with a huge heat release, resulting in fast strength development and large shrinkage, which can influence MPC on the repair efficiency of cement concrete pavement [13–16]. Many researchers found that the problem of short setting time with the huge reaction in heat of MPC is improved when either a cement clinker or CaO is mixed with retarders such as sodium tripolyphosphate (STP), borax ($Na_2B_4O_7 \cdot 10H_2O$), and boric acid ($H_3BO_3$) were used [17–19]. However, the addition of retarders may cause a reduction in mechanical property strength [20].

To improve the strength and durability of MPC, some modified materials and different kinds of fibers were introduced into MPC. Basalt fiber is generally considered as a potential material to improve the mechanical strength and toughness of the MPC. The flexural strength, post peak flexural performance, and fracture toughness of MPC with 0.5 percent to 1% basalt fiber is the best. In contrast, the mechanical property modification effect of basalt fiber in the MPC system is better than that of glass fiber [21–23]. The use of 0.05 weight percent graphene oxide as a promising nano-filler can improve the compressive and flexural strength of magnesium potassium phosphate cement (MKPC) pastes. At the same time, MPC pastes' early shrinkage strain was minimized [24,25].

Mineral admixtures were widely used in MPC systems to reduce material costs and improve the properties of MPC pastes. Adding ground granulated blast furnace slag (GGBS) to MPC mortar may increase its early mechanical performance due to the physical filler impact and the chemical reaction. It is worth noting that increasing GGBS fineness had lower effects on setting time and fluidity [26,27]. The addition of steel slag could promote the early hydration rate and water resistance of MPC pastes but had no positive effect on early strength development [28,29]. Silica fume (SF) is an active material with potential for application in the cement and concrete industries. According to previous experiments, a suitable proportion of SF improved the mechanical properties and water resistance of MPC. Moreover, the addition of SF increases the yield stress ($\tau_{paste}$) but decreases plastic viscosity ($\eta_{paste}$) slightly when increasing the SF content [30–32]. When using MPC as a construction repair material, it is often blended with fly ash (FA) and metakaolin (MK). Specimens of MPC that have been modified by dipotassium hydrogen phosphate have an increased expansion value when they contain a particular amount of FA [33]. With the incorporation of FA, MPC pastes show better water resistance but decrease the compressive strengths of MPC pastes [29,34]. It has been proven that alumina is an effective constituent in MK. In the MKPC system, MK is a more reactive component than FA, which makes MPC pastes exhibit higher compressive strengths [34,35]. However, the presence of MK has a detrimental impact on the fluidity of MPC, and the degree of the fluidity decrease has a direct link to the content of MK and its fineness [36].

At present, the research on MK and FA in MPC systems mainly focuses on the development of mechanical strength, workability and durability. However, the rheological properties and drying shrinkage of MPC materials mixed with FA and MK have not been systematic researched until now. As a vital factor, the rheological properties decide the performance of MPC. Proper rheological properties can ensure flowability and improve the stability of the paste. Low drying shrinkage can decrease MPC's volume change and reduce its cracking. In this study, the effects of varying doses of FA and MK as partial replacements for MgO on the workability, rheological characteristics, flexural strength, compressive strength, and drying shrinkage of MPC were examined. The possible mechanism of FA and MK in MPC has been discussed, which is beneficial for further promoting the design of MPC products.

## 2. Materials and Methods

### 2.1. Materials

Dead burnt MgO, ammonium dihydrogen phosphate ($NH_4H_2PO_4$, ADP), retarders, and mineral admixture are the basic components used to produce MPC pastes. MgO was obtained from Huanai Magnesia Materials Co. Ltd. (Yancheng, China), which has undergone a calcination process at a high temperature of 1600 °C. The ADP used in this study is industrial grade with a density of 1.801 g/cm³ and the PH value is 4.2–4.8. MgO was partially replaced with fly ash (FA) and metakaolin (MK). Table 1 summarizes the physical parameters and chemical composition of MgO, FA and MK. A laser particle size analyzer was used to examine the particle size distributions of dead burned MgO, FA and MK. The results of particle size analysis are displayed in Figure 1. It shows that the medium diameter of MgO is bigger than FA, and MK has the smallest particle size. In this research, new retarders were used, which were compounded by borax (B), sodium tripolyphosphate (STP), and sodium acetate anhydrous (SA) at a mass ratio of 6:1:1 ($m_B:m_{STP}:m_{SA} = 6:1:1$). The retarder to MgO mass ratio ($m_{retarder}/m_{MgO}$) was 0.16, and the purity of B, STP, and SA were all higher than 99%.

**Table 1.** Physical properties and chemical composition of raw materials (by wt/%).

| Code | Al$_2$O$_3$ | SiO$_2$ | CaO | MgO | Fe$_2$O$_3$ | TiO$_2$ | Na$_2$O | K$_2$O | Specific Surface Area (m²/kg) | Apparent Density (g/cm³) |
|------|-------------|---------|-----|-----|-------------|---------|---------|--------|-------------------------------|--------------------------|
| MgO | — | 2.09 | 1.31 | 92.15 | 1.12 | — | — | — | 920.6 | 3.48 |
| FA | 26 | 57.5 | 5.2 | 1.7 | 8.2 | 0.1 | 0.2 | 0.2 | 1121 | 1.99 |
| MK | 43 | 54 | 0.17 | 0.06 | 0.76 | 0.24 | 0.06 | 0.55 | 2613 | 2.23 |

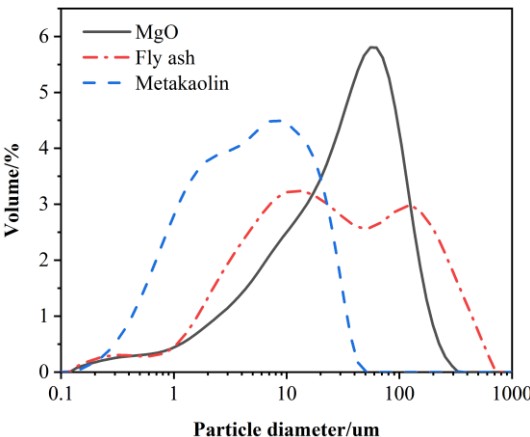

**Figure 1.** The particle diameters of MgO, fly ash and metakaolin.

### 2.2. Specimen Preparation

The magnesia to ADP mass ratio was found to be 4.0. In this study, the water to binder ratio was 0.17. Several contents of FA (0%, 5%, 10%, 15%) and MK (0%, 5%, 10%, 15%) were used to produce paste. Table 2 shows a total of 12 mixing proportions. MPC mixtures were prepared using a mechanical mortar mixer. The freshly mixed pastes were removed for workability and rheological property tests. After that, different MPC samples were made for the compressive strength test, the flexural strength test (40 mm × 40 mm × 160 mm) and the drying shrinkage test (25 mm × 25 mm × 280 mm). After 2 h of casting, the MPC samples were taken out of the mold and cured in corresponding standard experimental conditions 20 ± 2 °C temperature and more than 95% relative humidity.

**Table 2.** Mix proportions of MPC mortars containing FA and MK (by wt/%).

| Code | M | FA | MK | P/M [1] | W/C [2] | $M_{retarder}$/M [3] |
|---|---|---|---|---|---|---|
| Control MPC | 100 | 0 | 0 | 1/4 | 0.17 | 0.16 |
| FA5 | 95 | 5 | 0 | 1/4 | 0.17 | 0.16 |
| FA10 | 90 | 10 | 0 | 1/4 | 0.17 | 0.16 |
| FA15 | 85 | 15 | 0 | 1/4 | 0.17 | 0.16 |
| FA20 | 80 | 20 | 0 | 1/4 | 0.17 | 0.16 |
| MK5 | 95 | 0 | 5 | 1/4 | 0.17 | 0.16 |
| MK10 | 90 | 0 | 10 | 1/4 | 0.17 | 0.16 |
| MK15 | 85 | 0 | 15 | 1/4 | 0.17 | 0.16 |
| MK20 | 80 | 0 | 20 | 1/4 | 0.17 | 0.16 |
| FA5MK15 | 80 | 5 | 15 | 1/4 | 0.17 | 0.16 |
| FA10MK10 | 80 | 10 | 10 | 1/4 | 0.17 | 0.16 |
| FA15MK5 | 80 | 15 | 5 | 1/4 | 0.17 | 0.16 |

[1] M/P denotes the fixed mass ratio of MgO to $NH_4H_2PO_4$. [2] The mass ratio of water to binders is represented by W/C (binders include MgO, $NH_4H_2PO_4$, retarders, FA and MK). [3] $M_{retarder}$/M represents the mass ratio between retarders (retarders include B, STPP and SA) and MgO.

### 2.3. Testing Methods

(1)    Fluidity and setting time

The fresh properties of MPC were tested by conducting a fluidity test according to GB/T 8077-2012 [37]. First, the fresh MPC pastes were placed into a truncated tapered mold, and then the mold was lifted quickly. When there is no longer any development in the fluidity of pastes, a steel ruler was used to measure the diameter of the pastes from two perpendicular directions. To ensure the precision of the result, each fluidity test was recorded by a digital camera. The procedures for the tests are shown in Figure 2.

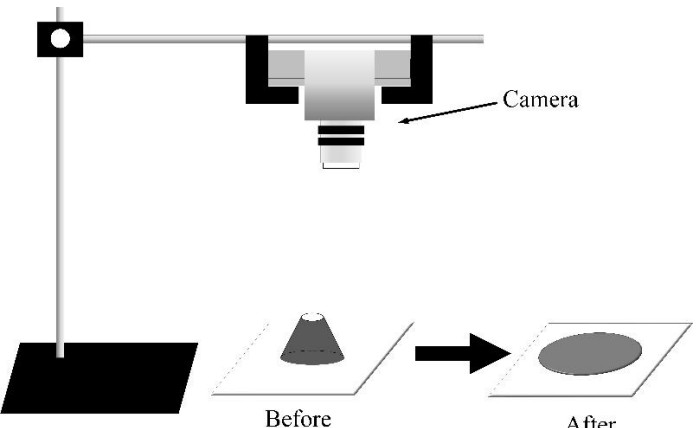

**Figure 2.** Schematic diagram of fluidity test.

A VICAT apparatus was utilized to measure the setting time of MPC, which was based on GB/T1346-2011 [38]. The time was recorded after adding water during the preparation process of MPC. When the time of the VICAT test needle was $4 \pm 1$ mm away from the bottom of the paste in the mold, the setting time of MPC material was determined.

(2)    rheological properties test

The fluidity is correlated to the yield stress ($\tau_{paste}$) of the paste, while the time it takes to obtain a prescribed spread is related to the viscosity ($\eta_{paste}$) of the paste, according to the relationship between the rheological characteristics of the paste and a fluidity test [39–41].

The procedures for the rheological tests are shown in Figure 2. A digital camera was used to record the process of the fluidity test. Based on the relationship between the fluidity and time which was recorded by the camera, the whole process of the MPC fluidity test

was analyzed. After determining the time for a specific paste flow diameter, the rheological parameters were calculated by Equations (1) and (2).

$$\tau_{paste} = \frac{225\rho_{paste}gV_{cone}^2}{128\pi^2(SF/2)^5} - \frac{0.005/\pi}{V_{cone}/\pi(SF/2)^2} \tag{1}$$

$$\eta_{paste} = \frac{\rho_{paste}ghV_{cone cone}}{150\pi \times slump \times SF_{pres}^2}t_{180} \tag{2}$$

where: $\rho_{paste}$—apparent density of MPC paste, $V_{cone}$—volume of paste in the truncated cone mold, $h_{cone}$—height of truncated cone mold, $SF_{pres}$—specific fluidity, *slump*—slump height at a specified fluidity, $T_{180}$—time when fluidity reaches 180 mm.

(3)　compressive and flexural strength

MPC paste cubes were examined at various curing ages of 1 h, 3 d, 7 d and 28 d, respectively. The Chinese standard GB/T 17671-1999 was referenced to conduct the flexural and compressive strength tests [42]. Two half specimens after the flexural strength test are used for the compressive strength test. Throughout the experiment, the load was determined at a rate of $2 \pm 0.4$ kN/s. The average compressive strength of three paste cubes was used in each test.

(4)　drying shrinkage

The drying shrinkage of MPC specimens at 1, 3, 7, 14, 21, 28, 60, 90, 180 days was measured using a comparator in which temperature and relative humidity of the curing condition are $20 \pm 3$ °C and $50 \pm 4$%, in accordance with JC/T 603—2004 [43]. The drying shrinkage of MPC specimens could be calculated by the following Equation (3).

$$\varepsilon = \frac{L_0 - L_t}{L_b} \times 100\% \tag{3}$$

where: $L_0$—length of specimen in 3 h curing age, $L_t$—length of specimen in *t* days curing age (t = 1 d, 3 d, 7 d, 14 d, 28 d, 60 d, 90 d, 180 d).

## 3. Results and Discussions

### 3.1. Workability

3.1.1. Effect of FA on MPC Fluidity and Setting Time

The effect of various FA contents on MPC fluidity and setting time is shown in Figure 3. It is noted that the fluidity of the paste increased with 5% FA first and then decreased with more FA. Compared with the control MPC, the fluidity increased by 3.5% and 1.2% with different FA content of 5% and 10%, respectively. However, the slump of MPC decreased by 3.9% and 6.7% when the FA amount is 15% and 20%. The incorporation of FA could improve the fresh mixture's workability due to its spherical particles' "ball-bearing"-type effect. However, when a large amount of FA is added, the FA particles need more free water to wrap, which reduces the amount in the free water of paste. It results in the poor workability of paste.

The setting time of the paste increases 29.6% with FA content rising from 0% to 15% and then decreases by 20.0% with FA content rising from 15% to 20%. The reason may be that the introduction of FA reduces the quality of MgO in the MPC system. However, the dosage of retarding components remains unchanged, and the ratio of retarder to MgO increases indirectly. Thus, the reaction rate and the hydration heat were reduced [44]. Furthermore, FA is an inert filler in MPC pastes. Adding FA can delay the hydration process and diminish the exothermic peak, resulting in a longer setting time for MPC pastes. However, when the range of FA is increased from 15% to 20%, the water absorption of FA plays a major role in reducing the water for the hydration reaction. In addition, the FA has adsorption characteristics. Excessive FA adsorbs the retarders, which restrict the retarding effect.

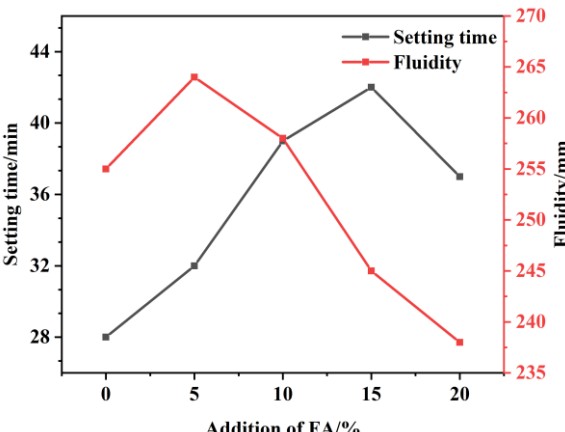

**Figure 3.** MPC pastes fluidity and setting time with varying fly ash contents.

### 3.1.2. Effect of MK on MPC Fluidity and Setting Time

Figure 4 shows that the addition of MK affects the fluidity and setting time of MPC pastes. The fluidity of the paste decreased 15.0% with MK content increasing from 0% to 20%. MK increases the density of cement and reduces the gap between cement particles due to its huge specific surface area and smaller particle size, resulting in poor paste fluidity. Furthermore, the MK particles are in irregular flakes, which can explain the decrease in fluidity after adding MK.

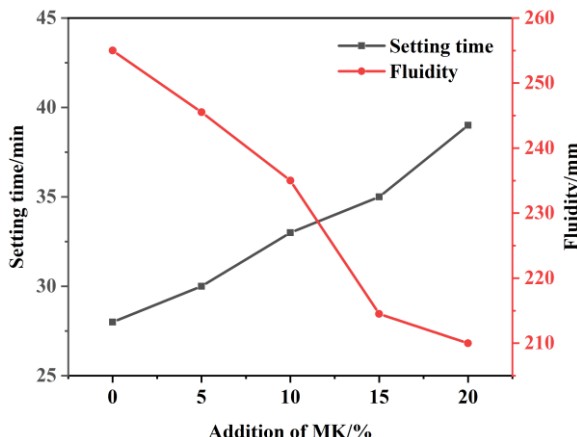

**Figure 4.** MPC pastes' fluidity and setting time with varying MK contents.

The setting time is increased with MK content ranging from 0% to 20%. The setting time of MPC extends to 39 min with 20% MK, which is nearly 40% longer than the case without MK. A possible reason for this is the formation of aluminum phosphate ($AlPO_4$) as a result of the reaction between the products from the dealumination of metakaolin ($Al^{3+}$) and the $PO_4$ units from ADP, which reduces the amount of MgO and the heat evolution of MPC.

### 3.1.3. Influence of Compound Powder on MPC Fluidity and Setting Time

The impact of various dosages of FA and MK on MPC setting time and fluidity is shown in Figure 5. In comparison to control MPC, the group contents of FA and MK show longer setting times and less fluidity. However, FA and MK have little influence on the fluidity when FA and MK were simultaneously applied to MPC in various ratios. As shown in Figure 5, considering the performance of fluidity and setting time comprehensively, the FA10MK10 sample has the best paste performance, while the fluidity of FA10MK10 was 225 mm, and the setting time of FA10MK10 was 45 min. The FA10MK10 sample

cannot only prolong the setting time but also have a good fluidity, which is beneficial to the construction.

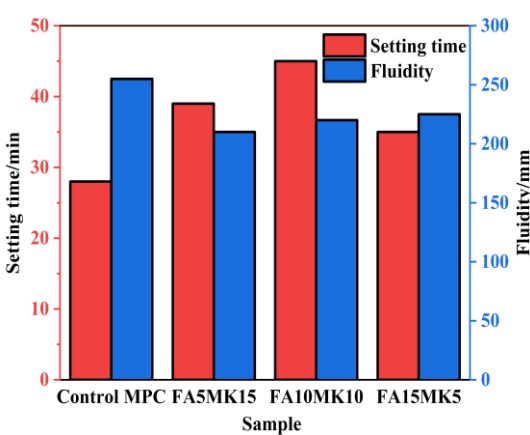

**Figure 5.** The fluidity and setting time of MPC pastes with compound powder.

*3.2. Rheological Properties*

3.2.1. Effects of FA on MPC Rheological Properties

The rheological property test results with different FA contents are shown in Figures 6 and 7. As illustrated in Figure 6, the T180 value of the paste decreases 13.0% as the FA content increases from 0% to 5%, and increases 28.0% as the FA content increases from 5% to 20%.

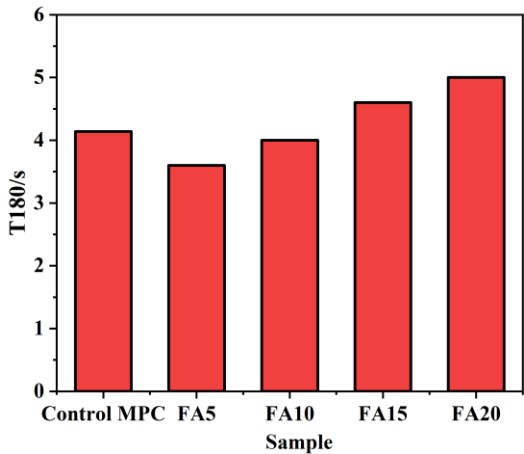

**Figure 6.** The effect of FA content on MPC T180 value.

The key parameters for evaluating the rheological properties of MPC pastes are yield stress ($\tau_{paste}$) and plastic viscosity ($\eta_{paste}$). The results are illustrated in Figure 7. The two indexes decrease significantly, with FA first decreasing and then increasing. Without the addition of FA, the yield stress and plastic viscosity of MPC were 1.41 Pa and 69.94 Pa·s, respectively, while the sample with 5% FA was 1.16 Pa and 60.08 Pa·s. After that, an increasing trend in yield stress and plastic viscosity began to occur. As observed in Figure 5, yield stress and plastic viscosity of samples with FA content ranging from 5% to 20% increased 38.6% and 25.7 percent, respectively.

It is generally thought that the rolling bearing effect of FA is the main reason for improving the fluidity of MPC. According to rheology research, the yield stress and plastic viscosity are reduced,. However, when the content of FA is more than 10%, the yield stress and plastic viscosity of the paste are increased. This can be interpreted as more FA particles needing a large amount of free water to wrap, reducing the paste's free water content.

On the other hand, FA reduced the volumetric weight of the MPC and thus increased the internal frictional force of the MPC.

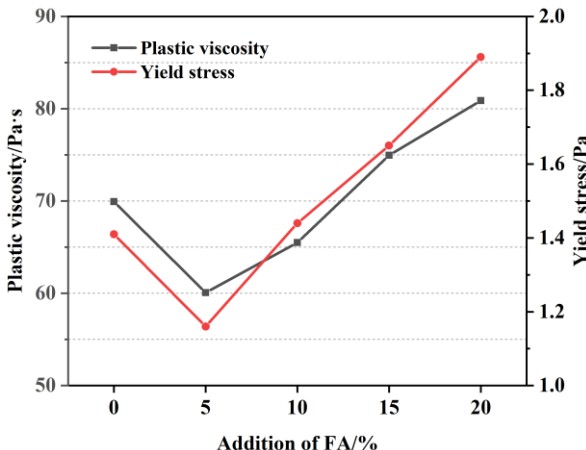

**Figure 7.** MPC rheological properties with different contents of FA.

### 3.2.2. Effects of MK on MPC Rheological Properties

Figures 8 and 9 show the rheological property test results of MPC incorporated MK. With an increase in MK content, the yield stress ($\tau_{paste}$) and plastic viscosity ($\eta_{paste}$) of MPC pastes gradually increase. As illustrated in Figure 8, the T180 value of control MPC is about 61.3% of MK20. It can be seen from Figure 9 that the yield stress of MPC containing 20% MK reached 3.56 Pa, which was 152% higher than the control MPC, while the MPC plastic viscosity reached 128.45 Pa·s, 84% higher than the control group. The irregular flake particle shape of MK increases the friction of fluidity. In addition, compared to MgO and $NH_4H_2PO_4$, MK has smaller particle sizes and larger specific surface areas. A higher specific surface area can reduce the free water content. The distance of particles will be close with MK content increase, leading to the increase in the friction forces of the particles.

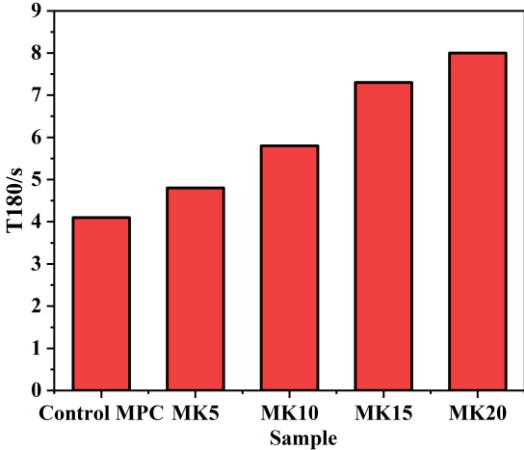

**Figure 8.** The effect of MK content on MPC T180 value.

### 3.2.3. Influence of Compound Powder on MPC Rheological Properties

Figure 10 shows the rheological properties test results of the MPC pastes containing various dosages of FA and MK. It is noted that the rheological parameters of MPC pastes with 20% composite mineral admixture content are greater than those of the contrast MPC pastes. Moreover, MK as a mineral additive in composites plays the leading role in the rheological properties of MPC. It could be seen from Figure 11 that the rheological parameter value of the FA5MK15 sample is maximal. The yield stress and plastic viscosity

of FA5MK15 were 3.24 Pa and 83.40 Pa·s, respectively. The compound powder with a large specific surface area and small particle size can effectively improve the viscosity.

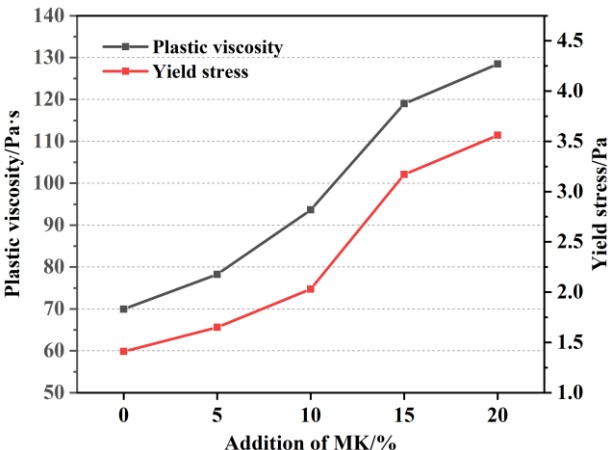

**Figure 9.** MPC rheological properties with different contents of MK.

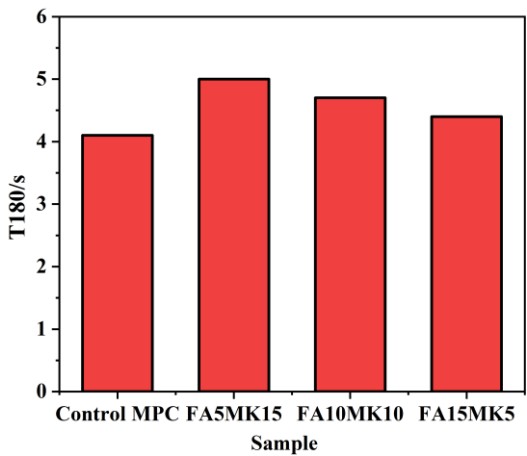

**Figure 10.** Influence of compound powder on T180 value of MPC.

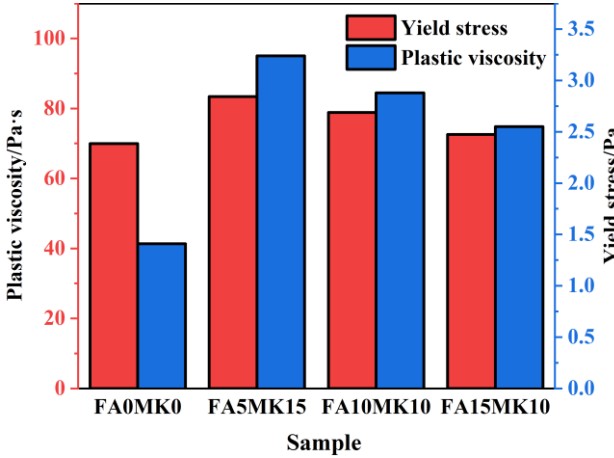

**Figure 11.** MPC rheological properties with addition of compound powder.

### 3.3. Mechanical Properties

### 3.3.1. Influence of Fly Ash on Mechanical Properties of MPC

Figures 12 and 13 show the mechanical properties test results of MPC with different FA contents. It can be seen from the fitting results that under different curing ages, the

fitting equations of each group are quadratic functions, and they show a downward trend. The flexural strength of all specimens increased with increasing curing time, and the 3 h flexural strength reached approximately seventy percent of the flexural strength at seven days. As shown in Figure 12, FA caused a slight flexural strength loss of the MPC sample. Compared with control MPC, the addition of 20% FA reduces the flexural strength of MPC from 6.3 to 5.8 MPa for 3 h, from 7.2 to 6.1 MPa for one day, and from 7.9 to 7.2 MPa for 28 days.

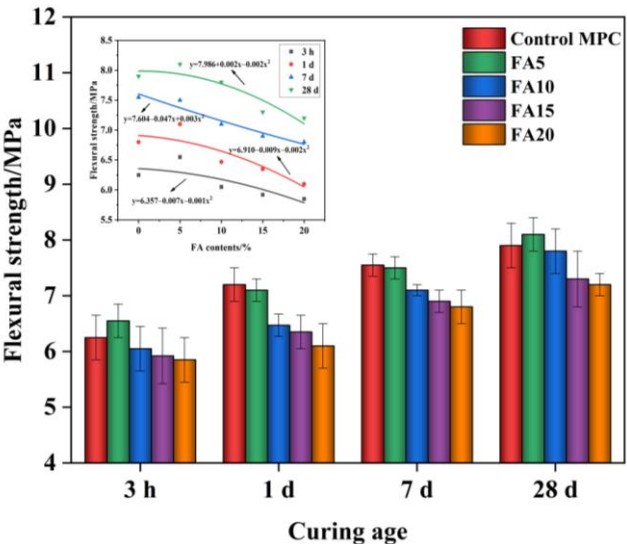

**Figure 12.** The effect of FA content on MPC flexural strength.

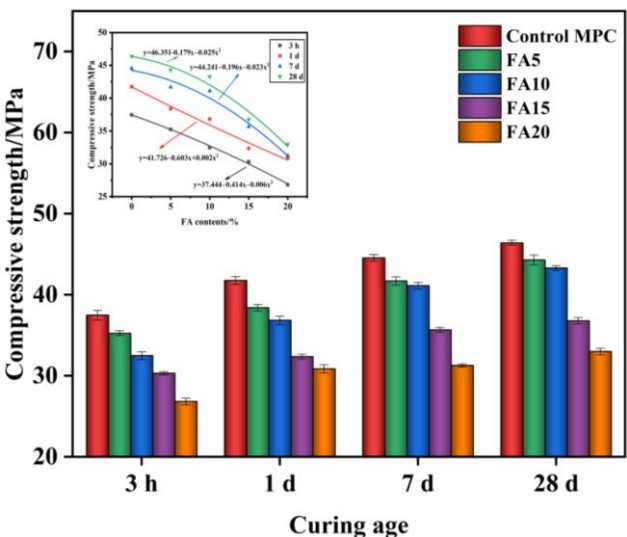

**Figure 13.** The effect of FA content on MPC compressive strength.

As illustrated in Figure 13, the control MPC compressive strength was higher than that of the other groups incorporating FA. The compressive strength decreases with FA content from 0% to 20%. It is worth noting that increasing the FA content from 0% to 10% does not result in a noticeable reduction in compressive strength. However, the compressive strength decreases by as much as 21% with FA content increasing from 10% to 20% after 28 days of curing. This result indicated that the presence of FA is averse to MPC strength due to the decreased MgO content, which reduced hydration products [34,45].

### 3.3.2. Influence of MK on MPC Mechanical Properties

Figures 14 and 15 reflect the impact of MK content on the flexural and compressive strengths of MPC pastes under standard curing conditions. The fitting results illustrated that under different curing ages, the fitting equations of each group are cubic functions, and they appear to first increase and then decrease. A similar trend between MPC compressive and flexural strength was observed when the MK content was increased. The mechanical properties improved with the MK content increase from 0% to 10% and decreased with the MK content increase from 10 to 20%. The MPC paste with 10% MK had the greatest compressive strength. MPC pastes containing 10% MK have a 28-day strength of over 50 MPa, nearly 12% higher than control MPC pastes. Meanwhile, the flexural strength of MPC pastes with MK reached 8.8 MPa after 28 days of curing, which is 11% greater than control MPC pastes. Although MK could participate in the reaction, enhance the compactness of the material and increase the strength, the optimization of MPC was reduced when adding MK to a percentage over 10%. On the one hand, MK has a larger specific surface area than MgO, which benefits the dense microstructure formation. On the other hand, MK can supply a large number of crystallization sites due to its high specific surface area. However, at a replacement rate over 10%, the mechanical properties decrease due to the decreasing hydration product [34–36].

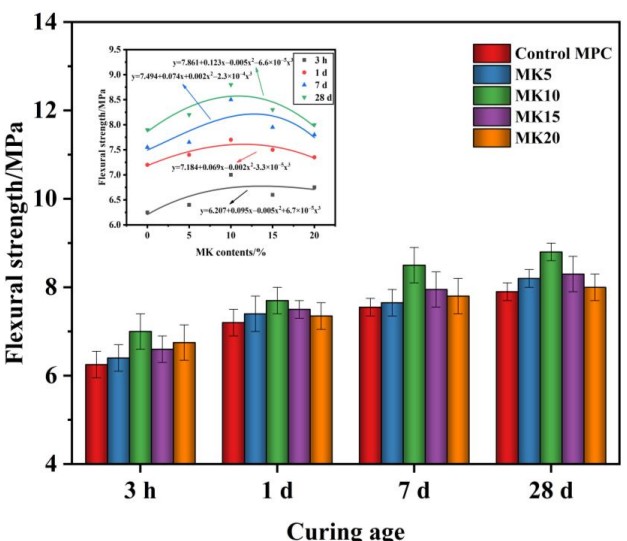

**Figure 14.** The effect of MK content on MPC flexural strength.

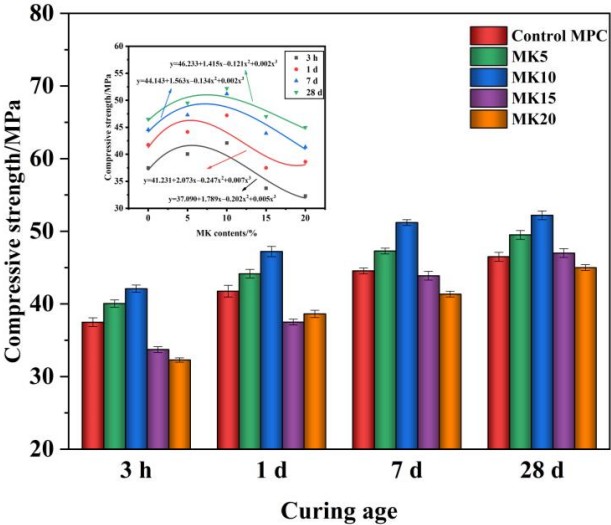

**Figure 15.** The effect of MK content on MPC compressive strength.

### 3.3.3. Influence of Compound Powder on Mechanical Properties of MPC

Figures 16 and 17 show the MPC mechanical properties containing various dosages of FA and MK. FA10MK10 indicated the best performance against flexural strength and compressive strength, with 10% FA content and 10% MK content.

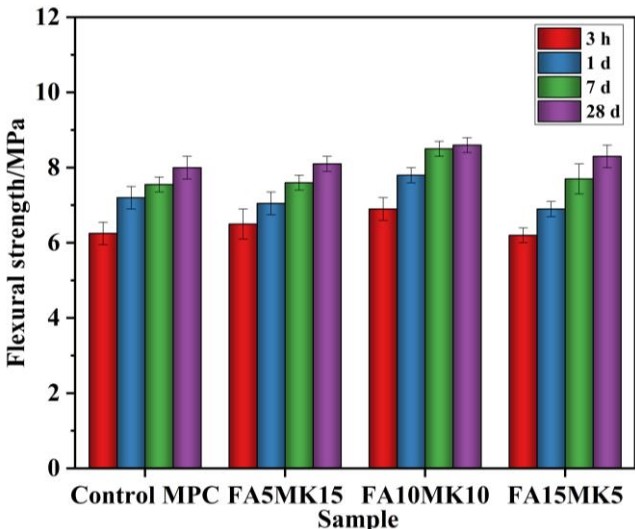

**Figure 16.** MPC flexural strength with addition of compound powder.

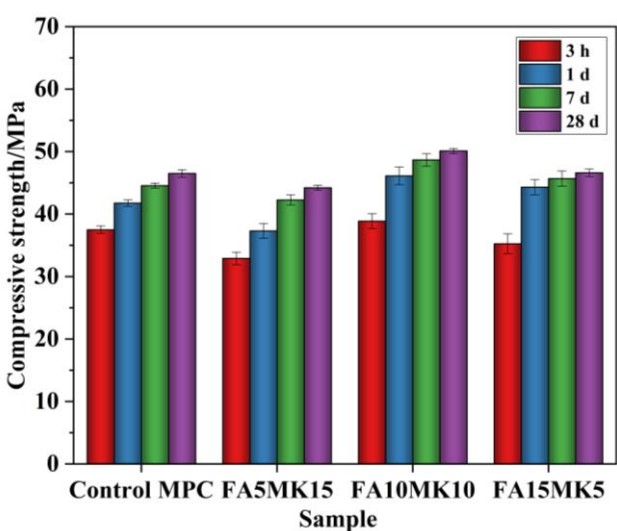

**Figure 17.** MPC compressive strength with the addition of compound powder.

The flexural strength of specimen FA10MK10 at 3 h, 1 day, 7 days and 28 days was 6.9 MPa, 7.7 MPa, 8.5 MPa and 8.6 MPa, which were 9.0%, 7.2%, 13.2% and 7.5% higher than that of control MPC, respectively. The FA10MK10 compressive strength at 3 h, 1 day, 7 days and 28 days was 39.1 MPa, 46.3 MPa, 48.9 MPa, and 50.1 Mpa, which were 4.0%, 10.1%, 8.9% and 7.7% higher than that of control MPC, respectively. The conclusion might be derived that adding MK and FA at the proper dosage could enhance mechanical strength, reduce cost and have remarkable economic benefits.

### 3.4. Drying Shrinkage

#### 3.4.1. The Effect of FA on MPC Drying Shrinkage

Figure 18 presents the drying shrinkage of MPC specimens with different contents of FA. The drying shrinkage of MPC specimens mainly happened within the first 14 days of curing. Compared with the control MPC, the shrinkage in F5, F10, F15, and F20 decreased

21.3%, 39.5%, 56.4%, and 73.8% at a curing age of 28 days, respectively. This evidence suggests that the addition of FA could effectively restrict MPC shrinkage. This can be explained by the reduction in hydration products due to the introduction of FA components. Moreover, the large surface tension of FA can be attributed to the improving MPC volume stability resulting from the rupturing of the MPC system bubble and filling the pores of the paste.

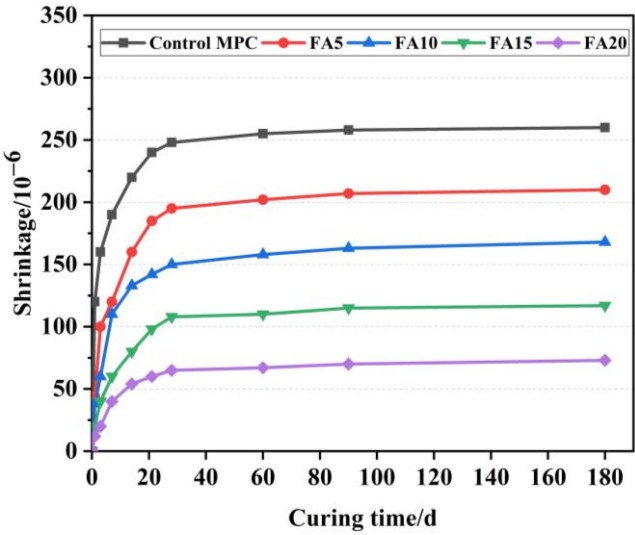

**Figure 18.** Effect of FA on MPC drying shrinkage deformation.

3.4.2. The Effect of FA on MPC Drying Shrinkage

Figure 19 illustrates the drying shrinkage of MPC specimens with different contents of MK. MPC drying shrinkage decreases as the content of MK increases. The drying shrinkage rate of specimens MK5 and MK20 at 28 days of curing is $120 \times 10^{-6}$ and $44 \times 10^{-6}$, which are only 48.4% and 17.7% that of the control MPC, respectively.

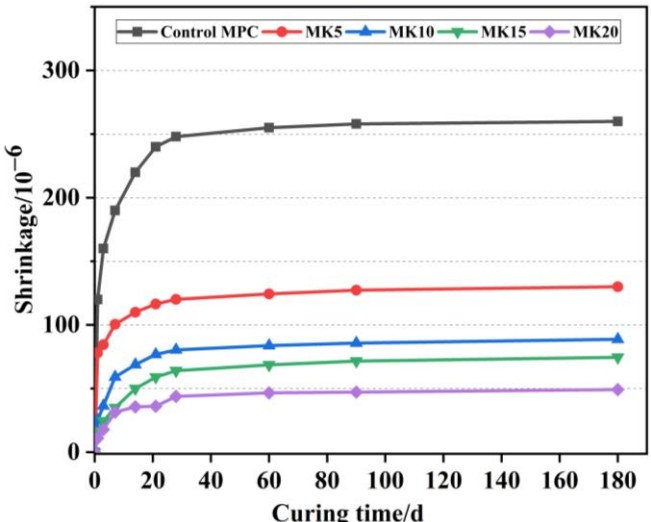

**Figure 19.** Effect of MK on MPC drying shrinkage deformation.

On one hand, MK has a smaller particle size, which could be filled with hydration products. This may be responsible for the significant shrinkage-free of MPC pastes. On the other hand, the flocculating substances formed after MK participates in the MPC hydration reaction are beneficial for optimizing the drying shrinkage of MPC pastes. Compared with FA, MK has a more considerable effect on MPC drying shrinkage optimization.

### 3.4.3. Influence of Compound Powder on Drying Shrinkage of MPC

Figure 20 illustrates the effect of compound powder consisting of FA and MK on the drying shrinkage of MPC pastes. The FA0MK0 sample was prepared without FA, and MK had the highest drying shrinkage value. The drying shrinkage of MPC pastes containing a 20% addition of mineral admixtures was less than $55 \times 10^{-6}$. The drying shrinkage rate of MPC decreases as the MK content of the composite addition increases. Among them, the minimum drying shrinkage rate of MPC was $36.5 \times 10^{-6}$, which occurred in the FA15MK5 sample. Hence, from the viewpoint of controlling shrinkage, when MPC composite paste with FA and MK, it has a more positive influence than conventional MPC products.

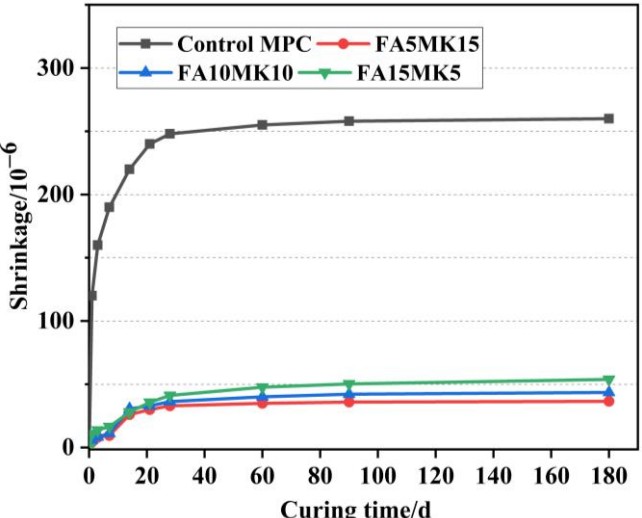

**Figure 20.** Drying shrinkage of MPC pastes with addition of compound powder.

### 3.5. Mechanism Analysis

Figure 21 shows the struvite crystal microstructure of MPC pastes without any mineral mixture addition after 28 days of curing. It grows into large tabular crystals rather than the initial short columnar and wedge-shaped crystals. At this time, the microstructure of MPC pastes is more compact. In addition, cracks could be seen in the hydration products of MPC pastes with higher magnification (Figure 21b). In some previous studies, it was considered that the dehydration of product, the internal stress caused by crystal growth and the effect of compression possibly caused the cracks [27,46].

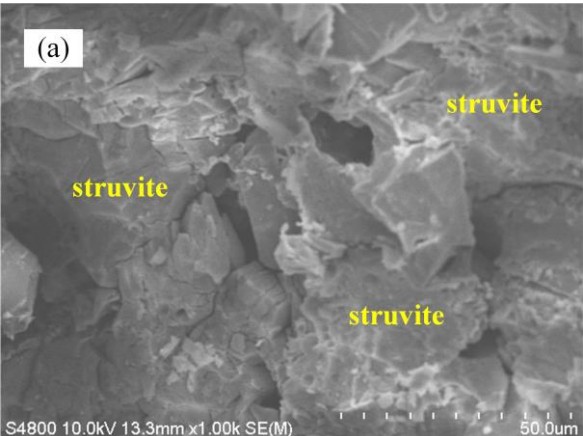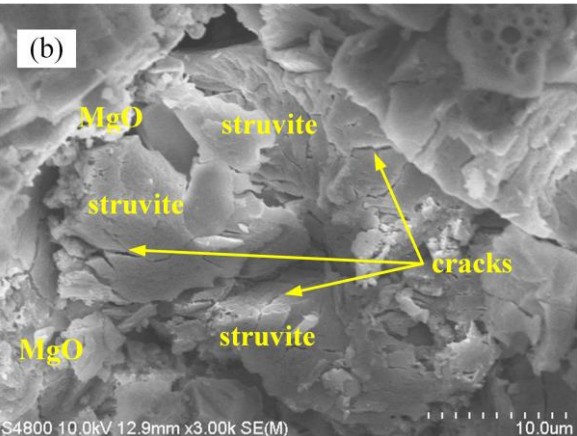

**Figure 21.** SEM microstructure of MPC at 28 d: (**a**) 1000 magnification, (**b**) 3000 magnification.

Figure 22 presents the microstructure of hydrated pastes containing 15 wt% FA by SEM at 28 days. It could be seen that the number of hydration products decreases when a large

amount of FA replaces MgO, which makes the hydration reaction incomplete. FA particles with smooth surfaces mainly exist in hydration products in round shapes. SEM images show that some FA particles are exposed on the surface of the matrix. Meanwhile, a clear interface could be observed between FA particles and hydration products. These could be confirmed FA is an inert filler in "conventional" designs of FA/MPC-based materials, its chemical activity could not be effectively stimulated [45]. Moreover, apparent gaps between the FA particles and the struvite are also observed in Figure 23b. The above will lead to poor compactness of the MPC microstructure and reduce the compressive strength.

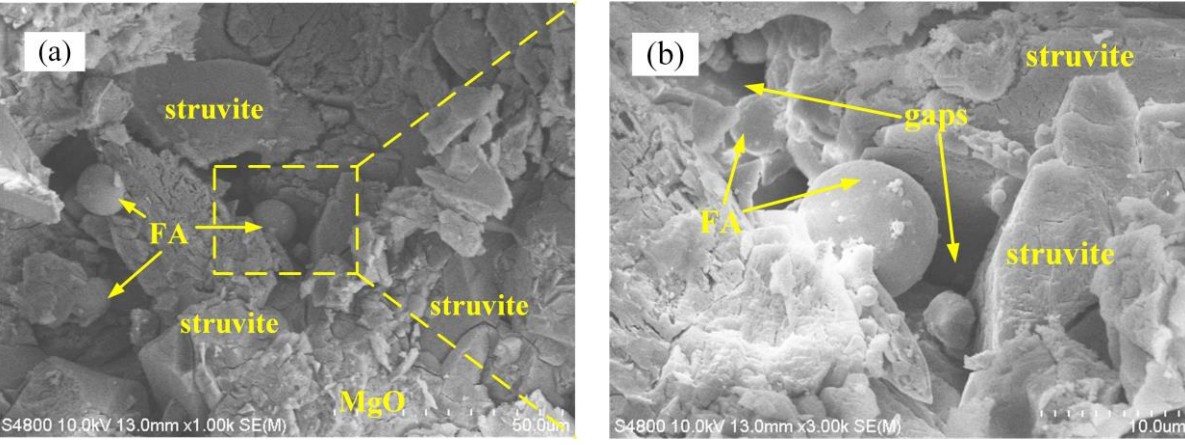

**Figure 22.** SEM microstructure of the MPC with the addition of FA at 28 d: (**a**) 1000 magnification, (**b**) 3000 magnification.

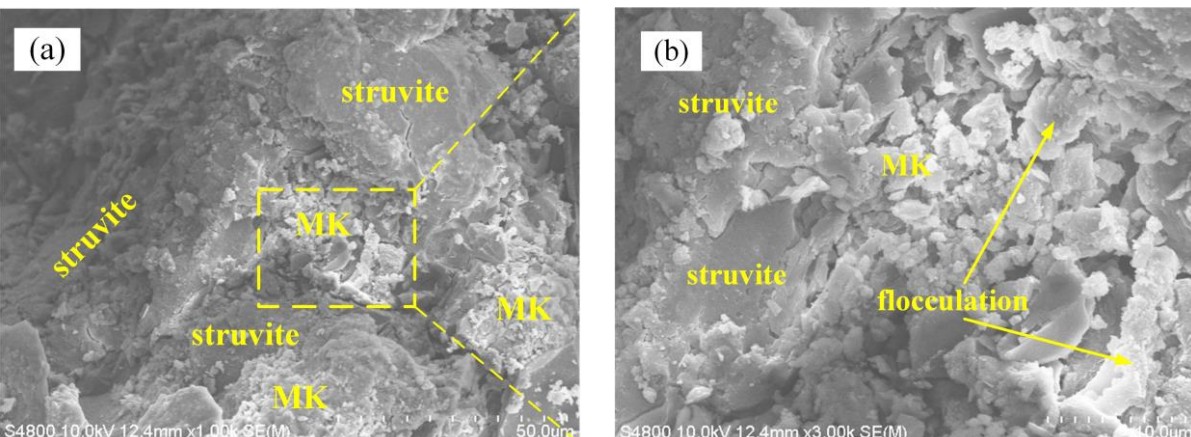

**Figure 23.** SEM microstructure of the MPC with the addition of MK at 28 d: (**a**) 1000 magnification, (**b**) 3000 magnification.

Figure 23 shows the microstructure of MPC pastes with the addition of MK at 28 d. The application of MK makes the porosity of MPC pastes lower, and less unreacted magnesia particles are noticed due to the possible pozzolanic reactions. From the SEM images, the formation of struvite clusters on the surface of MK is more homogenous. MK particles with smaller particle sizes were filled into the hydration products, which could significantly decrease the micro-cracks number. MK particles with smaller particle size were filled into the hydration products, which could decrease the number of micro-cracks significantly.

MK's filling effect promotes a more compact microstructure formed in MPC, and it can supply a large number of crystallization sites due to its high specific surface area. These are responsible for the significant shrinkage-free and mechanical properties optimization of MPC pastes.

Figure 24 shows the microstructure of MPC pastes containing 10 wt% MK and 10 wt% FA at 28 d. The composite addition of mineral admixtures consisting of FA and MK makes the MPC internal structure of the hardened body denser. It could be observed from Figure 24b that MK can effectively fill the gap between magnesium oxide particles and FA particles. The surface of some FA particles is covered by a certain amount of MK. This implies good chemical compatibility between FA and MK. Meanwhile, some unreacted magnesia particles were adsorbed by these two components. Mineral admixtures are beneficial to the optimization of the MPC microstructure.

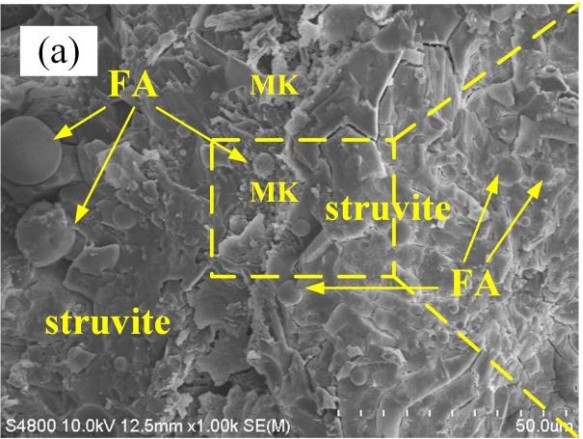 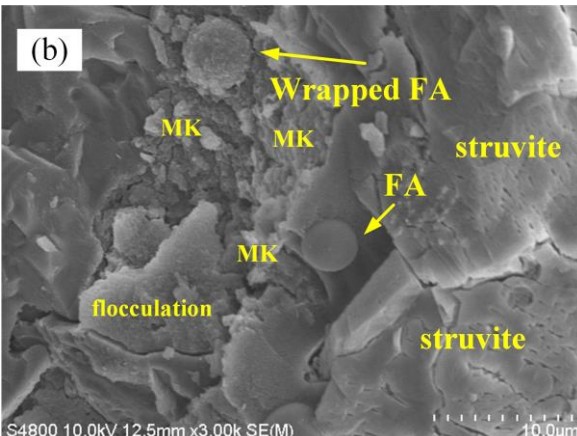

**Figure 24.** SEM images of the MPC pastes containing FA and MK at 28 d: (**a**) 1000 magnification, (**b**) 3000 magnification.

## 4. Conclusions

The effect of partial MgO substitution with FA and MK at various doses on workability, rheological properties, flexural strength, compressive strength, and the drying shrinkage of MPC was investigated in this study. The possible mechanism of FA and MK was analyzed through the microstructure of MPC. The main research results obtained in this study are as follows.

(1) MPC fluidity first showed an increasing trend and then decreased as the FA content varied from 0% to 20%, and the fluidity reached its best at 5% FA. The loss of MPC fluidity increases with increasing MK content continuously. With the increase of FA and MK, the setting time of MPC was prolonged. The replacement of 10 wt% FA and 10 wt% MK increases the setting time and decreases the fluidity by 36.9% and 15.5%, respectively.

(2) The yield stress and plastic viscosity first decreased and then increased with the increasing FA content in MPC. Furthermore, successive additions of MK increase the yield stress and plastic viscosity of MPC directly.

(3) The flexural and compressive strength of the MPC specimen decreases when the content of FA increases. In contrast, the presence of MK enhances the compressive and flexural strengths of MPC paste. Maximum compressive and flexural strength are attained in MPC paste with the 10% addition of MK. MPC with 10% MK and 10% FA exhibit both higher flexural and compressive strength than the samples without FA and MK at all test ages.

(4) Fly ash and MK have a positive influence on the shrinkage of MPC. The shrinkage rate decreases as the FA and MK content increase from 0 to 20%, respectively. It is notable that MK has a more considerable effect on MPC drying shrinkage optimization. MPC with 10% FA and 10% MK has the best-modified performance in terms of the comprehensive performance of MPC at all test ages and the shrinkage is small, which can be applied to rapid thin-layer mending material for cement concrete pavement.

(5) An SEM analysis showed that there were gaps between FA particles and struvites. This may be the main reason for the poor mechanical properties of MPC with different FA content. The incorporation of MK leads to a denser microstructure of the MPC mortars due to the filling effect and the formation of flocculating substances.

**Author Contributions:** Methodology, Funding acquisition, Project administration, Supervision, Writing—Original draft, H.L.; Formal analysis, Writing—Original draft, Q.F.; Supervision, Resources, Visualization, Y.Y.; Data curation, Software, J.Z.(Jingyi Zhang); Data curation, Software, J.Z.(Jian Zhang); Data curation, Writing—Original draft, Writing—Review and editing, G.D. All authors have read and agreed to the published version of the manuscript.

**Funding:** This work was supported by Doctoral Start-up Foundation of Liaoning province (grant numbers 2021-BS-166), Foundation of Liaoning Province Education Administration (grant numbers lnqn202017) and Open Fund of National Engineering Research Center of Highway Maintenance Technology (Changsha University of Science and Technology) grant numbers kfj210105.

**Institutional Review Board Statement:** Not applicable.

**Informed Consent Statement:** Not applicable.

**Data Availability Statement:** The data presented in this study are available on request from the corresponding author.

**Conflicts of Interest:** The authors declare no conflict to interest.

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
