# Peer review of "Experimental Research on Magnesium Phosphate Cements Modified by Fly Ash and Metakaolin"

_coatings, doi:10.3390/coatings12071030_

Round 1

Reviewer 1 Report

Reviewer comments on the paper ‘‘Experimental research on magnesium phosphate cements modified by fly ash and metakaolin” are below:

1.       Line 16, instead of magnesium phosphate cement you could write MPC as you stated before.

2.       Line number 34, Normally, the commonly used MgO is dead burned magnesium oxide. Please revise the statement, normally and commonly is the same thing.

3.       Line 146, 153, flexural and compressive strength tests Error! Reference source not found. Please fix the citation error.

4.       Line 351, Struvite crystals have developed completely after 28 days of curing. How did you know it? Please explain.

5.       Line 406, MPC fluidity first showed an increasing trend and then decreased as the FA content varied from 0% to 20%, and the fluidity reached its best at 5% FA. How did you explain the best of fluidity?

6.       Line 423, SEM analysis showed that there were gaps between FA particles and struvites. This may be the main reason for the poor mechanical properties, whose poor mechanical properties are?

7.       In general, the writing and presentation of the paper are poor, and need a significant revision to improve the quality of the paper, English is also not good enough for the publication, and many sentences do not make sense.

Reviewer 2 Report

First of all I wonder what that work has to do with Coatings. The work is a list of results, which do not demonstrate any study hypotheses, but appear to be the report of systematic tests of various types of additions to the cement to evaluate their effects.

However, the most critical paragraph is the one entitled Mechanism Analysis. We do not understand the mechanism of what, according to what it means "The possible mechanism of FA and MK 403 was analyzed through the microstructure of MPC."

The large amount of data collected by the authors should be better exploited, increasing the coherence between objectives and results.

Reviewer 3 Report

At the beginning of the text, I recommend addition of abbreviation names. For example, the meaning of MKPC, GGBS is not stated (although the term is generally known, I recommend adding it).

Line 59, sentence "Steel slag was a key byproduct of steel industry [24,25]." is completely unnecessary in the text, I recommend deleting.

In many cases, a text source error occurs in the text, so it is not possible to trace and check individual sources.

The introduction lacks a justification for the need for research.

The introduction is very brief, it should be expanded.

In the Materials chapter, only the properties of FA, MK and MgO are listed and described. Information on other components is not given.

Especially in chapter 3.3, the achieved results are only evaluated, the discussion is completely missing.

Overall, I consider the minimum discussion to be the weakest part of the article, which is otherwise very interesting and well organized.

In the end, there is no evaluation of whether it makes sense to add metakaolin or fly ash to the mixture and under what conditions, suggestions and recommendation.

Round 2

Reviewer 1 Report

The authors' response to reviewer is satisfactory, and can be considered for publication.

Reviewer 3 Report

The authors took my comments into account when revising the text. In my opinion, the article can now be published after minor editing of the text.